# Zerumbone Disturbs the Extracellular Matrix of Fluconazole-Resistant *Candida albicans* Biofilms

**DOI:** 10.3390/jof9050576

**Published:** 2023-05-16

**Authors:** César Augusto Abreu-Pereira, Ana Luiza Gorayb-Pereira, João Vinícius Menezes Noveletto, Cláudia Carolina Jordão, Ana Cláudia Pavarina

**Affiliations:** Department of Dental Materials and Prosthodontics, School of Dentistry, São Paulo State University (UNESP), Araraquara 14801-385, Brazil

**Keywords:** *Candida albicans*, phytotherapy, zerumbone, biofilms

## Abstract

This study assessed the effect of zerumbone (ZER) against fluconazole-resistant (CaR) and -susceptible *Candida albicans* (CaS) biofilms and verified the influence of ZER on extracellular matrix components. Initially, to determine the treatment conditions, the minimum inhibitory concentration (MIC), the minimum fungicidal concentration (MFC) and the survival curve were evaluated. Biofilms were formed for 48 h and exposed to ZER at concentrations of 128 and 256 µg/mL for 5, 10 and 20 min (*n* = 12). One group of biofilms did not receive the treatment in order to monitor the effects. The biofilms were evaluated to determine the microbial population (CFU/mL), and the extracellular matrix components (water-soluble polysaccharides (WSP), alkali-soluble polysaccharides (ASPs), proteins and extracellular DNA (eDNA), as well as the biomass (total and insoluble) were quantified. The MIC value of ZER for CaS was 256 μg/mL, and for CaR, it was 64 μg/mL. The survival curve and the MFC value coincided for CaS (256 μg/mL) and CaR (128 μg/mL). ZER reduced the cellular viability by 38.51% for CaS and by 36.99% for CaR. ZER at 256 µg/mL also reduced the total biomass (57%), insoluble biomass (45%), WSP (65%), proteins (18%) and eDNA (78%) of CaS biofilms. In addition, a reduction in insoluble biomass (13%), proteins (18%), WSP (65%), ASP (10%) and eDNA (23%) was also observed in the CaR biofilms. ZER was effective against fluconazole-resistant and -susceptible *C. albicans* biofilms and disturbed the extracellular matrix.

## 1. Introduction

Fungi are associated with several human diseases, ranging from superficial cutaneous and mucous infections to life-threatening systemic infections, depending on the host’s immunologic conditions [1]. *Candida albicans* was the most prevalent species in critically ill COVID-19 patients with oral candidiasis [2]. *C. albicans* may live in a mutualistic relationship with the host. However, this equilibrium can be lost under certain conditions, causing *Candida*-associated diseases [3,4]. In addition, the augmented resistance of *Candida* species to antifungal drugs is a serious healthcare issue, making research into alternative strategies against oral biofilms extremely relevant to public wellbeing [4,5].

Most *C. albicans* infections are associated with biofilm establishment on either biotic or abiotic surfaces, such as in protheses, catheters and implants [5]. Biofilms are complex microbial communities of adhered cells covered by an extracellular matrix that contributes to their preservation and to the maintenance of cells, surfaces and environmental interactions, hindering the action of conventional drugs [6,7]. The extracellular matrix of fungal biofilms is composed of polymers and extracellular DNA responsible for the biofilm structure’s maintenance [8]. Extracellular DNA plays a function in promoting the attachment of biofilm to the substrate [7,8]. Another important component present in the extracellular matrix is β-1,3-glucan, which plays a fundamental role in a biofilm’s resistance to conventional antifungals, since it prevents antifungal contact with target cells, resulting in the persistence or progression of the infection [9,10]. Therefore, one of the main challenges is treating infections caused by *C. albicans* biofilms since they present reduced susceptibility to conventional antifungals derived from azoles and polyenes [11,12]. In view of the increasing problem of drug resistance, essential oils can be considered a valuable resource due to their antimicrobial properties [11,13,14,15].

Essential oils are known for their analgesic, anti-inflammatory, antiviral, antioxidant, anticancer, immunomodulatory, antibacterial and antifungal properties [15]. Essential oils inhibit both the development and activity of *C. albicans* more efficiently than clotrimazole, and the damage caused by essential oils at the cellular level is stronger than that induced by antifungals [15]. Essential oils can affect the cell membranes of bacteria and fungi and can make them more permeable [16,17]. In general, antimicrobial properties are related to the interaction of essential oils and the cell system, especially against the plasma membrane and in the disturbance of mitochondrial functions [18].

Zerumbone (ZER) is a monocyclic sesquiterpene compound derived from the essential oil of *Zingiber zerumbet Smith*, which possesses several pharmacological properties, including antineoplastic, antioxidant, anti-inflammatory, immunomodulatory, antipyretic, antibacterial and antifungal properties [19,20,21,22,23,24]. ZER exhibits antimicrobial activity against *C. albicans*, *Candida tropicalis*, *Staphylococcus aureus*, *Staphylococcus epidermidis*, *Bacteroides fragilis*, *Acinetobacter baumannii* and *Escherichia coli* [24,25,26,27,28,29,30]. In addition, ZER ointment has a potent wound-healing capacity [25]. ZER inhibits the development of *C. albicans* biofilm by hindering hyphal growth, causing morphologic cell alterations [24] or ergosterol content reductions in fungal cell membranes [26].

Although the antimicrobial action of ZER against susceptible *C. albicans* has been evaluated previously [24,27,28], at this moment, there is no information about its effect on extracellular biofilm matrix components. Therefore, the present study aimed to assess the efficacy of ZER on biofilms of fluconazole-resistant (CaR) and -susceptible *C. albicans* (CaS). In addition, we evaluated the influence of ZER on extracellular matrix components: proteins, polysaccharides and eDNA.

## 2. Materials and Methods

### 2.1. ZER Solution Preparation

Stock solutions containing ZER (zerumbone, Sigma-Aldrich, St. Louis, MO, USA) were prepared prior to each experiment. ZER crystals were dissolved in 1% dimethyl sulfoxide (DMSO—Sigma-Aldrich, St. Louis, MO, USA) to achieve a final concentration of between 4 and 1024 µg/mL [24].

### 2.2. Culture of Microorganisms

*Candida albicans* strains (ATCC—American Type Culture Collection, Rockville, MD, USA) susceptible (ATCC 90028; CaS) and resistant (ATCC 96901; CaR) to fluconazole, which were stored in a −80 °C freezer, were thawed and reactivated on Petri plates containing Sabouraud dextrose agar (SDA—DIFCO Laboratories, Detroit, MI, USA) with chloramphenicol (50 mg/L), and then incubated at 37 °C for 48 h. To form the starter cultures, about five colonies of each strain were relocated to tubes containing 5 mL of yeast nitrogen base (YNB—DIFCO, Detroit, MI, USA) with glucose (100 mM) and incubated again at 37 °C for 16 h (pre-inoculum). At that time, 0.5 mL of pre-inoculum for each strain was transferred to correspondent tubes containing 9.5 mL of YNB in a 1:20 dilution. The inoculum was incubated at 37°C until it reached the optical density (OD) corresponding to the middle of the exponential growth phase (mid-log phase). The OD was determined at 540 nm: OD_540 nm_: 0.55 ± 0.08. Then, the CaS and CaR cultures were adjusted to obtain a microbial density of 10^7^ CFU/mL by washing the cells via centrifugation (4000× *g* for 5 min) and rinsing with PBS solution [31]. These procedures were performed three times.

### 2.3. Minimum Inhibition Concentration (MIC), Minimum Fungicidal Concentration (MFC) and Survival Curve

Testing susceptibility to ZER was achieved using minimal inhibitory concentration (MIC) and minimal fungicidal concentration (MFC) procedures. The Clinical and Laboratory Standards Institute microdilution method [32] was performed, with some changes, to evaluate the MIC. For control without *C. albicans*, 100 μL RPMI 1640 (2× concentrated) (Sigma-Aldrich, St. Louis, MO, USA) was used, buffered with MOPS (3-(N-morpholino) propanesulfonic acid) (Sigma-Aldrich, St. Louis, MO, USA) and 100 μL ultrapure sterile water. For growth control, CaS and CaR suspensions were evaluated without ZER. Minimal inhibitory concentration (MIC) and minimal fungicidal concentration (MFC) evaluations were performed using the Clinical and Laboratory Standards Institute microdilution procedure [32] to determine the concentration of ZER that would be used in the treatment. For this, ZER concentrations ranging from 4 to 1024 µg/mL were diluted in 1% DMSO [32]. For contamination control, 100 µL of RPMI 1640 (2× concentrated) (Sigma-Aldrich, St. Louis, MO, USA) and 100 µL of PBS were added (without fungal cells or ZER). For the ZER-free control, the inoculum was diluted in 2× concentrated RPMI (0.5 × 10^3^ to 2.5 × 10^3^ colonies forming unities/mL (CFU/mL)). For the positive control group, nystatin (Sigma-Aldrich, St. Louis, MO, USA) was tested in the range of 4 to 512 µg/mL. For the MIC test, the inoculum (0.5 × 10^3^ to 2.5 × 10^3^ UFC/mL) and the different concentrations of ZER tested (ranging 4–1024 µg/mL) were incubated in 96-well plates (Corning Co., Corning, NY, USA) with *C. albicans* suspension adjusted to obtain an inoculum concentration corresponding to 0.5 × 10^3^ to 2.5 × 10^3^ CFU/mL in RPMI. After incubation at 37 °C for 24 h, the plates were observed visually (for the presence or absence of growth) [32] and the optical density was measured (OD_492 nm_) using a plate reader (EZ Read 400 Microplate Reader, Biochrom^®^, Holliston, MA, USA). The MIC values were considered the lowest ZER concentration that caused a minimum of 50% decrease (MIC50) in growth related to the ZER-free growth control in the reading spectrophotometer of *C. albicans* culture [32]. These values were recorded in duplicate on three separate occasions.

In addition, plating and colony enumeration were performed to determine the MFC (i.e., the ZER concentration that causes the absence of fungal colony growth on agar plates after 24 h) and the survival curve at different times. For the survival curve, the plates were incubated with different concentrations of ZER for 5, 10, 20 and 30 min and 1, 2, 4, 6, 8 and 24 h. Then, plating and colony enumeration were performed to obtain colony-forming units (CFU/mL). The MFC was verified after 24 h of the biofilm’s treatment with ZER. The MIC was based on the growth density, and the lowest concentrations promoted a 50% reduction in the population compared to the ZER-free growth control. The MFC was considered the minimum concentration that resulted in the absence of fungal colony growth on agar plates after 24 h.

### 2.4. Biofilm Formation and Treatments

For the formation of CaS and CaR biofilms, 1 mL of CaS and Car inoculum was transferred individually to the wells of a 24-well polystyrene plate (K12-024; Kasvi, Sao José dos Pinhais, Paraná, Brazil) and next, 1 mL of Roswell Park Memorial Institute medium (RPMI 1640; Sigma-Aldrich, St. Louis, MO, USA) buffered with morpholine propane sulfonic acid (MOPS; Sigma-Aldrich, St. Louis, MO, USA) was added. Plates were incubated at 37 °C under orbital agitation for 90 min (75 rpm) to obtain microorganism adhesion. Following this, the wells were washed twice with 1 mL of PBS solution (8 g NaCl, 0.2 g KCl, 1.44 g Na2HPO4, 0.24 g KH2PO4, pH 7.4) to remove non-adhered cells and 1 mL of buffered RPMI (pH = 7) was added over again in each well and the plates remained in orbital agitation (37 °C/75 rpm). After 24 h of incubation, RPMI medium was removed by aspiration, 1 mL of fresh RPMI was added, and plates were incubated in agitation once again (37 °C/75 rpm) for another 24 h. After 48 h of biofilm formation, RPMI medium was removed, biofilms were washed with PBS twice [31], and the treatments were performed in three different groups: 1—control-group: biofilms did not receive any treatment; 2—ZER-128 group: biofilms were treated with ZER in a concentration of 128 µg/mL; and 3—ZER-256 group: biofilms were treated with ZER in a concentration of 256 µg/mL. In each experimental group, the treatments were performed for 5, 10 and 20 min. The experiments were performed in triplicate and on three different occasions (*n* = 12).

### 2.5. Efficacy Evaluation of ZER

At the end of the treatments, the biofilms were washed three times with NaCl (0.89%). After that, 2 mL of NaCl 0.89% was added to each well, and the biofilms were cautiously removed from the bottom of each plate with a pipette tip, transferred to sterilized microtubes, and submitted to sonication (30 s; 7 w; 190 J) [31]. After sonication, an aliquot (0.1 mL) of the suspension was separated into the enumeration of colony-forming units (CFU/mL) and another for total biomass determination (0.1 mL) [33,34]. The residual volume (1.8 mL) was centrifuged (5.500× *g*; 10 min; 4 °C), and the supernatant (1.8 mL) was divided into three aliquots for the analyses of soluble components of the matrix: water-soluble polysaccharides—WSP (1 mL) [31,35], eDNA (0.650 mL) [31,36] and proteins (0.150 mL) [37,38]. The pellet (insoluble components of extracellular matrix plus fungal cells) resulting from centrifugation was resuspended in Mili-Q water and divided into different aliquots for insoluble biomass quantification (0.8 mL) [31,34] for insoluble protein quantification (0.05 mL) [34,37] and for the determination of alkali-insoluble polysaccharides—ASP (0.95 mL) [31,35].

### 2.6. Statistical Analyses

The normal distribution and homoscedasticity of data for each matrix component (dependent variable) were analyzed using Shapiro–Wilk and Levene tests, respectively. Then, variance analysis (two-way ANOVA), followed by a Bonferroni post-test, was used in order to verify interactions among the treatment factors (concentration of ZER and time of exposure; independent variables). The significance level adopted was 5% (α = 0.05). The analyses were carried out using SPSS software (IBM^®^ SPSS^®^ Statistics, version 29, Chicago, IL, USA) with a significance level of 5% (α = 0.05).

## 3. Results

### 3.1. MIC, MFC and Survival Curve

The MIC50 values of ZER observed in the susceptibility test were 64 µg/mL for CaR and 256 µg/mL for CaS. For the OD_492 nm_ mean value of the control (the ZER-free growth control), the CaS mean was 0.756 ± 0.081, and the CaR mean was 0.686 ± 0.081. A concentration of 256 μg/mL was the lowest concentration of ZER that promoted a 50% reduction (0.318 ± 0.062) in CaS. A concentration of 64 μg/mL promoted a 50% reduction (0.301 ± 0.036) in CaR. For the positive control (nystatin), the MIC values observed were 8 μg/mL (OD_492 nm_ 0.167 ± 0.062) for CaS and 512 μg/mL (OD_492 nm_ = 0.249 ± 0.002) for CaR.

The survival curve demonstrated that 256 µg/mL ZER after 2 h promoted a 99% reduction in viable colonies for CaS and CaR compared to the initial inoculum (Figure 1). For CaR, 128 µg/mL ZER exhibited fungicidal activity after 2 h. The total reduction in fungal growth after 24 h (256 and 128 µg/mL for CaS and CaR, respectively) was equivalent to the ZER concentrations that promoted a reduction in the count of viable colonies as a function of time. The MIC value observed for CaR (64 µg/mL) was not similar to the MFC. At a concentration of 64 µg/mL, a reduction occurred only in the first 10 min (a 55.31% reduction), and after this time, an increase in the number of viable colonies was observed (Figure 1). Thus, the concentration of 128 µg/mL was the choice for the next CaR experiments since it promoted a reduction of more than 99% in the viability of CaR.

### 3.2. Efficacy of ZER on Biofilm Components

The interactions between the treatment factors are described in Table 1. Means and standard deviations for biofilm components are described in Table 2 and Table 3 for CaS and CaR, respectively.

For the CFU/mL of CaS and CaR, an interaction (*p* ≤ 0.001) between the time of exposure and the ZER concentrations evaluated was observed (Table 1). For all evaluated times, the ZER-256 group presented the greatest reduction in biofilm viability, being statistically different from the ZER-128 and control groups of CaS (*p* ≤ 0.009) and CaR (*p* ≤ 0.001). Evaluating the ZER concentrations during the evaluated times, it could be observed that the ZER-256 group showed no significant differences between 10 and 20 min (*p* = 1.000) for CaS (Table 2); however, for CaR, a significant reduction in cell viability was observed at 20 min (*p* ≤ 0.003) (Table 3).

The total dry weight results showed no interaction between the time and concentration evaluated (*p* ≥ 0.959) (Table 1). Analyzing the treatment factors for CaS separately, there was an interaction with the concentration (*p* < 0.001). The ZER-256 group presented a significant reduction in total dry weight values and was different from the other groups (*p* ≤ 0.006) at all evaluated times. The ZER-128 group showed a lower reduction compared to ZER-256; however, it was different from the control group (*p* < 0.001) (Table 2). The ZER treatment evaluations did not change the total dry weight values of CaR biofilm (*p* ≥ 0.089) (Table 3).

For insoluble weight, no interaction was observed among the treatment factors for CaS and CaR (*p* ≥ 0.101). Then, the factors were evaluated separately, and the ZER concentration factor showed some interaction (*p* < 0.001) (Table 1). For the two evaluated strains, the ZER-256 group presented a significant reduction in dry weight values and was different from the other groups (*p* ≤ 0.033) at all of the evaluated times. For CaS, the ZER-128 group was statistically different from the control group at 5 and 10 min (*p* ≤ 0.029) (Table 2). For CaR, the ZER-256 group presented a major reduction in insoluble weight and was different from the other groups (*p* ≤ 0.001) (Table 3).

The results of soluble proteins showed no interaction among treatment factors for CaS (*p* = 0.122); however, this interaction was observed for CaR (*p* ≤ 0.001). Then, the factors were analyzed separately for CaS, and interaction was observed in the time factor (*p* < 0.001) (Table 1). For CaS, the ZER-256 group was different from the control group (*p* = 0.004) at 5 min (*p* = 0.004); however, it behaved similarly to the ZER-128 group (*p* = 0.747) (Table 2). The groups were statistically similar at 10 and 20 min (*p* = 1.000). For CaR, the ZER-256 and ZER-128 groups showed significant reductions at 10 and 20 min (*p* ≤ 0.001) compared with the control group and were statistically similar (*p* = 1.000) (Table 3).

For insoluble proteins, an interaction was observed among the treatment factors for CaS (*p* = 0.014) and CaR (*p* = 0.047) (Table 1). For CaR, no significant difference was observed among the groups when analyzing them separately (*p* ≥ 0.053). On the other hand, the ZER-256 group presented a significant reduction in insoluble proteins at 20 min compared with the other groups (*p* ≤ 0.009) (Table 3).

The results of the WSP showed interaction among the treatment factors for both strains (*p* ≤ 0.040) (Table 1). For CaS, the ZER-128 group showed a significant reduction after 10 min and was different at 5 min (*p* = 0.012) and similar at 20 min (*p* = 1.000). The ZER-256 group exhibited a major reduction in WSP compared to the other groups (*p* ≤ 0.001) and was similar at all evaluated times (*p* = 1.000). After 10 min of treatment, the ZER-128 group showed a reduction in WSP amount compared to the control group (*p* < 0.001); however, on a smaller scale than the ZER-256 group (Table 2). For CaR, the ZER-256 group showed a major reduction in WSP levels, which were statistically different from the others (*p* ≤ 0.001) (Table 3).

The results of ASP showed no interaction among the treatment factors (*p* = 0.916) for CaS. However, for CaR, an interaction was observed among the treatment factors (*p* = 0.021) (Table 1). The ZER-256 group presented a reduction in ASP levels at 10 and 20 min, which was statistically different from the control group (*p* = 0.044 for CaS (Table 2) and *p* = 0.037 for CaR (Table 3)).

In the eDNA analyses, interaction was observed among the treatment factors for CaS (*p* = 0.001) and CaR (*p* = 0.002) (Table 1). For both evaluated strains, the ZER-256 group presented a significant reduction in eDNA values compared to the other groups (*p* ≤ 0.001 for CaS (Table 2) and *p* ≤ 0.013 for CaR (Table 3)). In addition, the ZER-128 group was also statistically different from the control group (*p* < 0.001 for CaS and *p* < 0.032 for CaR).

## 4. Discussion

Natural chemical compounds that present activated biomolecules with antimicrobial action have become a promising alternative to the inactivation of microorganisms resistant to conventional antimicrobials [39,40,41,42]. The antioxidant and antibacterial capabilities of essential oils are well documented; however, studies on antifungal activities are still limited. From a health point of view, finding effective and safe antifungal agents to control the growth of *Candida* spp. is important. Recently, some studies have demonstrated the antibiofilm activities of ZER against Gram-positive and Gram-negative bacteria [24,26,27,28,29,30]. However, the effect of ZER on the extracellular matrix components of *C. albicans* biofilms has not yet been fully elucidated. Thus, this study investigated whether ZER interferes with the extracellular matrix of fluconazole-susceptible and -resistant *C. albicans* biofilms. The results demonstrated that ZER significantly reduced the cell viability and extracellular matrix components (WSP, ASP, eDNA, proteins) of fluconazole-susceptible and -resistant *C. albicans* biofilm.

The present study demonstrated strong ZER antifungal activity against CaR and CaS, with MIC of 64 μg/mL and 256 μg/m, respectively. These results corroborate a previous study that detected MIC in the range 64–128 μg/mL for methicillin-resistant *Staphylococcus aureus* strains (SA1199B, ATCC25923, XU212, RN4220 and EMRSA15) and 250 μg/mL for *Streptococcus mutans* [42]. In another previous study, fluconazole-resistant (ATCC 96901) and -susceptible *C. albicans* (ATCC 90028) strains showed fluconazole MIC values of 256 μg/mL and 16 μg/mL, respectively [33]. ZER presents an extensive variety of biological actions, with high therapeutic potential and antimicrobial activity [25,27,43,44,45]. ZER is a monocyclic sesquiterpene that is the major component of *Zingiber zerumbet* Smith essential oil [46]. Terpenoids act on specific phases of the *C. albicans* cellular cycle, inhibiting and interfering with cell adhesion [24,28]. These substances promote changes in membrane permeability and fluidity, resulting in cell wall degradation, which also affects fungal adhesion [45]. Moreover, these components act like inhibitors of morphogenesis from yeasts to hyphal, and when they are added to biofilms in the initial phase, they prevent the evolution of biofilms [46].

The cell viability of the CaS and CaR biofilms was reduced by approximately 37% when treated with ZER at concentrations of 256 µg/mL. These biofilms presented a 17% reduction of viability when treated with ZER at concentrations of 128 µg/mL. In a previous study, it was observed that ZER, at a concentration of 256 µg/mL, promoted a more than 50% reduction in the metabolic activity of fluconazole-susceptible *C. albicans* (CaS) biofilms (ATCC 14053 and two clinical isolates) [24,28]. In addition, ZER inhibited the adhesion of cells to surfaces and the maturation of preformed biofilms in a dose-dependent mode [24,28].

The extracellular matrix of biofilm consists of an extensive array of functional biomolecules such as exopolysaccharides (β-glucans, α-mannans), nucleic acids (eDNA), proteins, lipids and other biomolecules [7]. The extracellular matrix serves as a scaffold for structural support and a dynamic environment that provides varying chemical and physical signals to microbial communities, promoting biofilm existence [7,10,47]. When biofilms are already established, approaches that can reduce extracellular polymeric substances may dismantle the scaffolding/protective matrix, weaken the biofilm’s structure and potentiate antimicrobial killing. Considering the structural organization of the extracellular matrix of biofilm, the results of the present study were very promising since ZER reduced the polysaccharides (WSP and ASP) and extracellular DNA (eDNA) of fluconazole-susceptible and -resistant *C. albicans* biofilms’ extracellular matrixes. The results showed that CaS and CaR biofilms had their WSP reduced by 60% and 65%, respectively, when treated with ZER at concentrations of 256 µg/mL, regardless of the time of application. In addition, a reduction of approximately 10% in the amount of ASP in both evaluated strains was noticed after 20 min of application. In a previous study, it was observed that sublethal concentrations of *Perilla frutescens* essential oil also promoted a WSP reduction of approximately 80% from the extracellular matrixes of fungal biofilms [48]. On the other hand, the WSP levels of biofilms from fluconazole-susceptible *C. albicans* were not affected after treatment with alternative antifungal therapies [31,49].

*Candida albicans* biofilms are structured by the mannans—glucan complex (MGCx), formed by the interaction between WSPs (α-mannans) and ASPs (β-glucans) [10]. The integrity of the MGCx and its relations are fundamental elements of the antifungal resistance noticed in *Candida* biofilms [47]. The mannan-1,6-glucan conjugate is the major matrix constituent, while in the cell wall, β-1,3 glucan is the predominant cell wall polysaccharide [50,51]. In addition, the biofilm presents increased β-1,3 glucan content in *C. albicans* cell walls compared to planktonic organisms, making the biofilm more resistant to conventional antifungal therapies [9,10]. β-1,3-glucan secreted by *C. albicans* prevents the penetration of antibacterial drugs, providing enhanced antimicrobial protection for *S. aureus* within mixed biofilms [9,47]. On the other hand, *S. mutans* glucans surrounding *Candida* cells directly bind and sequester antifungal agents, reducing drug uptake and enhancing *C. albicans* tolerance within mixed biofilms [9,50]. This way, the reduction in polysaccharides (WSP and ASP) is a sign that ZER essential oil can actuate, promoting the disorganization of MGCx interactions, favoring antimicrobial activity.

Beyond polysaccharides (ASPs and WSPs), the MGCx also contains extracellular DNA molecules (eDNA) so that these components are interconnected and participate in the structural maintenance of the biofilm [8]. eDNA may interact with diverse extracellular polymeric substances, contributing to the biofilm’s structural organization, serving as a nutrient source, while promoting protection against antimicrobials, horizontal gene transfer and surface adhesion [52]. In *C. albicans* biofilms, both the polysaccharide matrix (WSP and ASP) and eDNA have demonstrated contributions to antifungal drug tolerance [7,10]. In the present study, ZER reduced the eDNA present in biofilm by approximately 75% for CaS and 23% for CaR. This eDNA reduction, promoted by ZER, may be related to the weakening of the *C. albicans* biofilm matrix, since the eDNA performs functions essential to biofilm formation, tending to the maintenance of structural integrity and inducing the morphological transition from yeast to hyphal during development [51,52,53].

The total biomass of biofilm consists of cells and the soluble and insoluble components of MEC [31,34]. The results revealed that ZER reduced the total biomass of the CaS biofilms in a dose-dependent manner, showing a reduction of 40% for 128 µg/mL and 55% for 256 µg/mL. On the other hand, the total biomass of the CaR biofilms was not altered by the treatments performed. In a previous study using confocal laser microscopy-specific markers, it was also noticed that 64 µg/mL of ZER promoted a reduction in the total biomass and cellular density of mixed *C. albicans* (ATCC 14053) and *S. aureus* (ATCC 14053) biofilms [28]. In addition, 128 µg/mL of ZER led to the mixed biofilm’s structural degradation [28]. CaS and CaR biofilms had their insoluble biomass reduced by approximately 44% and 13%, respectively, when treated with ZER at concentrations of 256 µg/mL for 20 min. Similar results were observed with antifungals (nystatin and amphotericin B), which promoted a dose-dependent reduction in the insoluble biomass of *C. albicans* biofilms [54].

Proteins are components present on a large scale in biofilm [10,11]. In the present study, no reduction was observed in soluble proteins of CaS and CaR biofilms. On the other hand, the ZER-256 group reduced the insoluble proteins in the CaR and CaS biofilms by approximately 15% after 10 and 20 min, respectively. In a previous study, a reduction in *C. albicans* biofilm proteins after alternative antifungal therapies was not observed, even with a reduction in cellular viability and other components of the extracellular matrix [31,55]. The reduction in proteins could be an important approach since they play an important role in the biofilm’s dynamic, acting like a digestive microstructure that performs the rupture of extracellular biopolymers in order to obtain energy [51].

The antibiofilm potential of ZER is not restricted to fungal biofilms, since the substance affects other microorganisms, such as *Staphylococcus aureus* [28], *Bacteroides fragilis* [29] and *Acinetobacter baumannii* [28,30]. Furthermore, ZER has also shown low cytotoxicity in mammal cells [56]. Thus, ZER can be considered a promising alternative to the inactivation of mixed biofilms.

## 5. Conclusions

Extracellular polymeric substances can act as antifungal diffusion-limiting barriers, resulting in restricted drug contact with the cells in the deeper layers of the biofilm [57]. The effect of ZER against vital constituents of the extracellular matrix (WSP, ASP and eDNA) can be considered a very relevant result that can improve the delivery of antifungals and could affect the antifungal resistance of the biofilm. In addition, the exposure of established biofilms to ZER reduced cell viability and decreased the amounts of eDNA, WSPs and the insoluble dry weight of biofilms from fluconazole-resistant *C. albicans*. This represents a promising alternative approach to antibiofilm therapy that requires further investigation of in vivo models.

## Figures and Tables

**Figure 1 jof-09-00576-f001:**
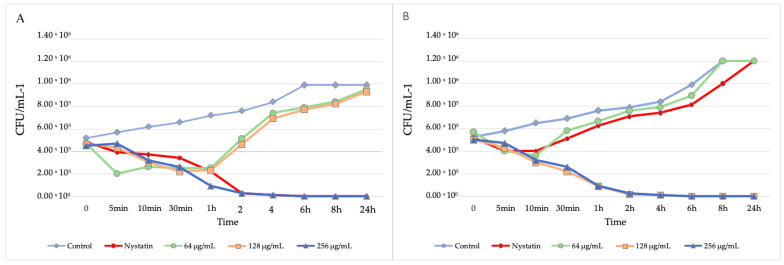
Survival curve of CaS (**A**) and CaR (**B**) after ZER treatment. The data represent the mean values of viable colony count (CFU/mL^−1^) at different concentrations of ZER and the times evaluated. The nystatin was evaluated as a positive control. The standard deviation (SD) was not higher than 1.71 × 10^5^ and it was not presented in the figure for clearer presentation of the data.

**Table 1 jof-09-00576-t001:** Summary of the results of the interactions of the two treatment factors (concentration of ZER and time of exposure) for each biofilm component and strain evaluated (CaS and CaR). Significant values are shown in bold (α = 0.05).

Component	Strain	Factor		Interation
		Concentration	Time	Concentration versus Time
CFU/mL	CaS	***p* < 0.001**	***p* < 0.001**	***p* < 0.001**
	CaR	***p* = 0.004**	***p* = 0.012**	***p* < 0.001**
Dry-Weight (mg)	CaS	***p* < 0.001**	*p* = 0.092	*p* = 0.959
	CaR	*p* = 0.860	*p* = 0.936	*p* = 0.992
Insoluble Dry-weight (mg)	CaS	***p* < 0.001**	*p* = 0.150	*p* = 0.257
	CaR	***p* < 0.001**	***p* = 0.013**	*p* = 0.101
Soluble proteins (µg)	CaS	*p* = 0.081	***p* < 0.001**	*p* = 0.122
	CaR	***p* < 0.001**	***p* < 0.001**	***p* < 0.001**
Insoluble proteins (µg)	CaS	*p* = 0.089	*p* = 0.078	***p* = 0.014**
	CaR	*p* = 0.698	*p* = 0.424	***p* = 0.047**
WSP (µg)	CaS	***p* < 0.001**	*p* = 0.089	***p* = 0.040**
	CaR	***p* < 0.001**	*p* = 0.051	***p* = 0.039**
ASP (µg)	CaS	***p* < 0.001**	*p* = 0.737	*p* = 0.916
	CaR	***p* < 0.001**	*p* = 0.069	***p* = 0.021**
eDNA (ng)	CaS	***p* < 0.001**	*p* = 0.253	***p* = 0.001**
	CaR	***p* < 0.001**	***p* < 0.001**	***p* = 0.002**

**Table 2 jof-09-00576-t002:** Biofilm and extracellular matrix components of CaS biofilms after ZER treatments.

Component	Groups	5 min		10 min		20 min	
		Mean	Standard Deviation	Mean	Standard Deviation	Mean	Standard Deviation
CFU/mL	Control	8.33 × 10^5 Aa^	1.71 × 10^5^	1.16 × 10^6 Ba^	1.24 × 10^5^	2.17 × 10^6 Ca^	2.77 × 10^5^
	ZER-128	1.63 × 10^5 Ab^	4.96 × 10^4^	2.40 × 10^5 Bb^	7.82 × 10^4^	4.13 × 10^5 Cb^	1.30 × 10^5^
	ZER-256	1.10 × 10^5 Ac^	3.02 × 10^4^	1.73 × 10^5 Bc^	5.21 × 10^4^	1.73 × 10^5 Bc^	3.94 × 10^4^
Dry-Weight (mg)	Control	5.733 ^Aa^	0.939	5.266 ^Aa^	0.695	5.566 ^Aa^	0.752
	ZER-128	3.400 ^Ab^	0.898	3.066 ^Ab^	0.574	3.383 ^Ab^	0.829
	ZER-256	2.349 ^Ac^	0.383	2.166 ^Ac^	0.389	2.450 ^Ac^	0.566
Insoluble Dry-weight (mg)	Control	1.200 ^Aa^	0.062	1.198 ^Aa^	0.053	1.128 ^Aa^	0.115
	ZER-128	1.119 ^Ab^	0.099	1.096 ^Ab^	0.085	1.110 ^Aa^	0.055
	ZER-256	0.663 ^Ac^	0.035	0.641 ^Ac^	0.069	0.624 ^Ab^	0.060
Soluble proteins (µg)	Control	0.024 ^Aa^	0.002	0.022 ^Ba^	0.001	0.021 ^Ba^	0.001
	ZER-128	0.023 ^Aab^	0.002	0.022 ^ABa^	0.001	0.021 ^Ba^	0.001
	ZER-256	0.022 ^Ab^	0.002	0.022 ^Aa^	0.001	0.021 ^Aa^	0.001
Insoluble proteins (µg)	Control	0.010 ^Aa^	0.001	0.010 ^ABa^	0.002	0.011 ^Ba^	0.002
	ZER-128	0.010 ^Aa^	0.001	0.010 ^Aa^	0.001	0.011 ^Aa^	0.002
	ZER-256	0.010 ^Aa^	0.001	0.010 ^Aa^	0.001	0.009 ^Ab^	0.001
WSP (µg)	Control	0.090 ^Aa^	0.009	0.092 ^Aa^	0.014	0.091 ^Aa^	0.004
	ZER-128	0.083 ^Aa^	0.006	0.071 ^Bb^	0.017	0.069 ^Bb^	0.017
	ZER-256	0.033 ^Ab^	0.003	0.032 ^Ac^	0.002	0.031 ^Ac^	0.001
ASP (µg)	Control	0.100 ^Aa^	0.008	0.101 ^Aa^	0.009	0.100 ^Aa^	0.006
	ZER-128	0.100 ^Aa^	0.012	0.096 ^Aab^	0.002	0.094 ^Aab^	0.009
	ZER-256	0.092 ^Aa^	0.010	0.094 ^Ab^	0.002	0.092 ^Ab^	0.007
eDNA (ng)	Control	35.904 ^ABa^	2.99	35.189 ^Aa^	4.52	39.356 ^Ba^	6.05
	ZER-128	22.452 ^Ab^	4.00	22.167 ^Ab^	3.82	21.669 ^Ab^	5.27
	ZER-256	14.903 ^Ac^	1.05	11.720 ^ABc^	2.54	8.541 ^Bc^	1.78

The data are shown as average and standard deviation (*n* = 12): viable colonies counting (CFU/mL); Dry-Weight (mg); Insoluble Dry-Weight (mg); Soluble proteins (µg); Insoluble proteins (µg); Water-soluble polysaccharides (WSP; µg); Alkali-soluble polysaccharides (ASP; µg); extracellular DNA (eDNA; ng). Capital letters show the comparison among the times (lines) and lowercase letters show the comparison among the concentrations (columns). Unequal letters indicate significant statistical differences (*p* < 0.05).

**Table 3 jof-09-00576-t003:** Biofilm and extracellular matrix components of CaR biofilms after ZER treatment.

Component	Groups	5 min		10 min		20 min	
		Mean	Standard Deviation	Mean	Standard Deviation	Mean	Standard Deviation
CFU/mL	Control	4.91 × 10^6 Aa^	5.13 × 10^5^	4.97 × 10^6 Aa^	3.92 × 10^5^	4.85 × 10^6 Aa^	5.02 × 10^5^
	ZER-128	2.63 × 10^6 Ab^	1.37 × 10^5^	2.27 × 10^6 Bb^	2.05 × 10^5^	2.51 × 10^6 Cb^	2.27 × 10^5^
	ZER-256	1.61 × 10^6 Ac^	2.52 × 10^5^	1.07 × 10^6 Bc^	1.72 × 10^5^	5.73 × 10^5 Cc^	3.94 × 10^4^
Dry-Weight (mg)	Control	4.967 ^Aa^	0.496	5.000 ^Aa^	0.572	5.033 ^Aa^	0.450
	ZER-128	4.950 ^Aa^	0.444	4.900 ^Aa^	0.357	4.950 ^Aa^	0.683
	ZER-256	4.900 ^Aa^	0.463	5.000 ^Aa^	0.621	4.967 ^Aa^	0.558
Insoluble Dry-weight (mg)	Control	1.211 ^Aa^	0.066	1.196 ^Aa^	0.049	1.179 ^Aa^	0.073
	ZER-128	1.131 ^Ab^	0.071	1.123 ^Ab^	0.057	1.131 ^Aa^	0.045
	ZER-256	1.108 ^Ab^	0.049	1.069 ^ABb^	0.050	1.016 ^Bb^	0.054
Soluble proteins (µg)	Control	0.024 ^Aa^	0.001	0.026 ^Aa^	0.002	0.025 ^Aa^	0.002
	ZER-128	0.024 ^Aa^	0.001	0.022 ^Bb^	0.001	0.021 ^Bb^	0.001
	ZER-256	0.024 ^Aa^	0.002	0.021 ^Bb^	0.001	0.021 ^Bb^	0.001
Insoluble proteins (µg)	Control	0.010 ^Aa^	0.001	0.010 ^Aa^	0.001	0.010 ^Aa^	0.001
	ZER-128	0.010 ^Aa^	0.000	0.010 ^Aa^	0.001	0.010 ^Aa^	0.001
	ZER-256	0.010 ^Aa^	0.001	0.010 ^Aa^	0.001	0.010 ^Aa^	0.001
WSP (µg)	Control	0.090 ^Aa^	0.009	0.093 ^Aa^	0.014	0.092 ^Aa^	0.004
	ZER-128	0.084 ^Ab^	0.006	0.072 ^Ab^	0.017	0.070 ^Ab^	0.017
	ZER-256	0.053 ^Ac^	0.003	0.053 ^Ac^	0.002	0.051 ^Ac^	0.001
ASP (µg)	Control	0.098 ^Aa^	0.005	0.098 ^Aa^	0.005	0.099 ^Aa^	0.004
	ZER-128	0.096 ^Aa^	0.005	0.096 ^Aab^	0.004	0.095 ^Ab^	0.002
	ZER-256	0.096 ^Aa^	0.004	0.094 ^Ab^	0.003	0.089 ^Bc^	0.002
eDNA (ng)	Control	61.571 ^Aa^	3.387	61.523 ^Aa^	3.473	60.940 ^Aa^	2.715
	ZER-128	59.119 ^Aa^	3.596	58.000 ^Ab^	4.478	55.003 ^Bb^	3.251
	ZER-256	55.320 ^Ab^	2.976	51.889 ^Bc^	2.324	46.708 ^Cc^	3.143

The data are shown as average and standard deviation (*n* = 12): viable colonies counting (CFU/mL); Dry-Weight (mg); Insoluble Dry-Weight (mg); Soluble proteins (µg); Insoluble proteins (µg); Water-soluble polysaccharides (WSP; µg); Alkali-soluble polysaccharides (ASP; µg); extracellular DNA (eDNA; ng). Capital letters show the comparison among the times (lines) and lowercase letters show the comparison among the concentrations (columns). Unequal letters indicate significant statistical differences (*p* < 0.05).

## Data Availability

Additional data are available on request from the corresponding authors.

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
