# Peer review of "Zerumbone Disturbs the Extracellular Matrix of Fluconazole-Resistant Candida albicans Biofilms"

_jof, 2023, doi:10.3390/jof9050576_

Round 1

Reviewer 1 Report

Major issue

ATCC 90028 and ATCC 96901 are not isogenic strains. The Authors should not discuss their results in terms of the effect of Zerumbone regarding drug resistance of C. albicans if both strains are of a different isogenic line.

Fig 1. SD should be included within the chart. If SD bars affects readability of the chart an appriopriate comment should be included in the captions, such as the SD was ommitted due to clearer presenting of the results including the informations that SD was not higher than...

Also in Tab 1 and 2 all the parameters should be quantified in triplicate and SD should be included. 

Minor issues:

Text editting, examples of mistakes: degree C with different symbol; commas instead of stops (example 0,24 g); 2x insted of 2×. In some parts of the manuscript the Authors use "h" in others "hours" - it should be unified. β-glucans instead of β-glucanos). "spp." should not be italicized. ETC.

Author Response

Reviewer 1 comments and suggestions for Authors:

  • ATCC 90028 and ATCC 96901 are not isogenic strains. The Authors should not discuss their results in terms of the effect of Zerumbone regarding drug resistance of albicans if both strains are of a different isogenic line.

Thank you for the input. The strains evaluated are not isogenic. In the present study, it was evaluated the effect of ZER against fluconazole-resistant and -susceptible Candida albicans strains. C. albicans can acquire resistance to fluconazole in various ways: by diminishing the toxic effects of the drug, by altering the target enzyme to reduce drug binding, by increasing the amount of the target enzyme, or by preventing the intracellular accumulation of the drug. All these mechanisms have been found in fluconazole-resistant clinical C. albicans isolates (Morschhäuser, 2016). In the present study, the aim was to evaluate the effect of ZER in strains with different characterizes. In a previous study, it was observed that the minimal inhibitory concentration to fluconazole for C. albicans ATCC96901 (CaR) was 256 μg/mL and for C. albicans ATCC90028 (CaS) was 16 μg/mL (Panariello et al., 2018). Here, we discuss the results considering the antimicrobial action of ZER between the strains comparing with the results obtained in a previous study.

References:

Morschhäuser J. The development of fluconazole resistance in Candida albicans - an example of microevolution of a fungal pathogen. J Microbiol. 2016;54(3):192-201. doi:10.1007/s12275-016-5628-4.

Panariello, B.; Klein, M. I.; Mima, E.; & Pavarina, A. C. Fluconazole impacts the extracellular matrix of flucona-zole-susceptible and -resistant Candida albicans and Candida glabrata biofilms. Journal of Oral Microbiology. 2018, 10, 1476644.

  • Fig 1. SD should be included within the chart. If SD bars affects readability of the chart an appriopriate comment should be included in the captions, such as the SD was ommitted due to clearer presenting of the results including the information that SD was not higher than...

Thank you for the input. The standard deviation (SD) was not higher than 1.71E+05 and it was not presented in the figure for clearer presenting the data. Figure 1 was modified to improve the visualization of the data.

  • Also in Tab 1 and 2 all the parameters should be quantified in triplicate and SD should be include

The experiments were performed in triplicate and in three different occasions (n=12). The means and standard deviation in triplicate were showed in Tables 1 and 2.

  • Minor issues:Text editting, examples of mistakes: degree C with different symbol; commas instead of stops (example 0,24 g); 2x insted of 2×. In some parts of the manuscript the Authors use "h" in others "hours" - it should be unified. β-glucans instead of β-glucanos). "spp." should not be italicized. ETC.

The manuscript was revised and corrected. 

Reviewer 2 Report

In the paper entitled "Zerumbone disturbs the extracellular matrix of fluconazole-resistant Candida albicans biofilms", the authors present the antifungal effect of zerumbone (ZER) extracted from the essential oil of Zingiber zerumbet Smith and which is the major substance in ginger.

The idea of the research is interesting, but the paper is written in a very bad English which made it very difficult to review. It clearly needs some major changes before acceptance in JoF:

- Major English language editing by an native/certified English language speaker;

- there are many doubled or missing words (the authors should check/read carefully their manuscript before submitting it);

- MIC and MFC values should be provided (figure 1 is not really relevant si difficult to follow since the authors have used no color, just shades of grey); in addition, the antifungal activity evaluation should also include an antifungal reference drug;

- "in vivo"/"in vitro" should be in Italic face.

Author Response

Reviewer 2 comments and suggestions for Authors:

  • In the paper entitled "Zerumbone disturbs the extracellular matrix of fluconazole-resistant Candida albicans biofilms", the authors present the antifungal effect of zerumbone (ZER) extracted from the essential oil of Zingiber zerumbet Smith and which is the major substance in ginger. The idea of the research is interesting, but the paper is written in a very bad English which made it very difficult to review. It clearly needs some major changes before acceptance in JoF:

Thank you for the comments. The manuscript was improved as suggested.

  • Major English language editing by an native/certified English language speaker.

The manuscript was submitted to the English language editing and the certificate of revision was attached.

  • There are many doubled or missing words (the authors should check/read carefully their manuscript before submitting it).

The manuscript was improved as suggested.

  • MIC and MFC values should be provided (figure 1 is not really relevant si difficult to follow since the authors have used no color, just shades of grey); in addition, the antifungal activity evaluation should also include an antifungal reference drug;

The antifungal activity of Nystatin was included in the manuscript and the data was presented in Figure1. The figure was improved for better visualization.

  • "in vivo"/"in vitro" should be in Italic face.

The manuscript was improved as suggested and the corrections were made.

Reviewer 3 Report

I have attached a file for the authors.

Author Response

The manuscript by Abreu-Pereira and co-workers is very interesting and well planned. However, there are some issues that need to be clarified and corrected in the manuscript, as follows:

Line 48 – Neither ref. 9 or ref. 10 show that beta glucan in the extracellular matrix prevents antifungal activity.

Thanks for the comment. References have been updated.

Nett J, Lincoln L, Marchillo K, et al. Putative role of β1,3 glucans in Candida albicans biofilm resistance. Antimicrob Agents Chemother. 2007;51:510–520

Taff HT, Nett JE, Zarnowski R, et al. A Candida biofilm-induced pathway for matrix glucan delivery: implications for drug resistance. PLoS Pathog. 2012;8(8): e1002848.

Line 82 – Instead of “Reactivation” use the word “Culture”

The manuscript was revised.

Line 87 – Why using chloramphenicol to grow candida? It is well known that chloramphenicol acts negatively in yeasts! See these publications (DOI: 10.1016/j.mycmed.2014.10.019; doi:https://doi.org/10.1167/iovs.17-22047 )

The chloramphenicol was added to the culture medium to inhibit bacterial growth, allowing isolated Candidagrowth. Chloramphenicol is a broad-spectrum antibiotic that inhibits protein synthesis in bacterial cells. It is often used in microbiology to prevent bacterial growth during the cultivation of other microorganisms such as fungi. When growing Candida species, it is important to ensure that there is no bacterial contamination as bacteria can compete with Candidafor nutrients, altering the test results and impairing the interpretation of results. Thus, chloramphenicol is added to the culture medium to inhibit bacterial growth (Murray et al.,2011).

Murray PR, Baron EJ, Jorgensen JH, Landry ML, Pfaller MA. Manual of Clinical Microbiology. 10th ed. Washington, DC: American Society of Microbiology Press; 2011.

Joseph MRP et al. (DOI: 10.1016/j.mycmed.2014.10.019) observed that the strains of Candida albicans were resistant to chloramphenicol (200 mg/mL). Three out of the four tested Candida albicans as well as other species (Candida famata, Candida glabrata, Candida haemolonei and Cryptococcus neoformans) showed no inhibition zones to chloramphenicol (200 mg/mL) and were considered resistant to this agent at this concentration. Blanco et al. (DOI:10.1167/ iovs.17-22047) found that isolated chloramphenicol has weak, or any, antifungal activity against several Candida yeasts relating to fluconazol. Thus, chloramphenicol can be added to the culture medium to inhibit bacterial growth without acting negatively in yeasts.

Line 158 – How did you determine biomass? Only by dry weight?

The total biomass was determined with 100 µL of the sample that were pipetted and placed on previously weighed and labeled aluminum paper, as well as 100 µL of the 0.89% NaCl solution used on each occasion. The samples were dried in an oven at 100°C, and weighed again on a precision balance. The difference between the initial weight and the final weight was calculated and then the value of the weight of the 0.89% NaCl solution was unconsidered. The final value was considered the total dry weight of the sample.

For insoluble weight, a 0.95 mL aliquot of the biofilm suspension was immediately placed on ice. After the aliquot was thawed, the tube was centrifuged (13000 xg/10 min/4°C). The supernatant was carefully removed from the tube and discarded. Then, the precipitate was dried in a sample concentrator for approximately 3 h. The precipitates obtained were weighed on an analytical balance to determine the insoluble dry weight.

Koo H, Hayacibara MF, Schobel BD, et al. Inhibition of Streptococcus mutans biofilm accumulation and polysaccharide production by apigenin and tt-farnesol. J Antimicrob Chemother. 2003;52:782–789.

Panariello, B.H.D.; Klein, M.I.; Pavarina, A.C.; Duarte, S. Inactivation of genes TEC1 and EFG1 in Candida albicans influences extracellular matrix composition and biofilm morphology. J. Oral. Microbiol 2017, 9, 1385372. [CrossRef] [PubMed] 47.

Line 175 - In the Results section please indicate the figure and Table where it can be found the results.

The manuscript was corrected and the Figures and Tables were indicated.

Line 179 and throughout the entire document – use “.” instead of “,” to indicate decimal place

The manuscript was corrected.

Line 197 and remaining Figures – the figure caption should have a title instead of “Mean values and …”

The figures captions were rewrite.

Line 200 – What’s the meaning of “MEC's components”?

The writing was changed for “Efficacy of ZER on biofilm components”.

Line 209 – It’s better to include a table with the statistical study regarding these relations between the period of exposure and the other variables. It is too confusing to read the table as it is…

A table with statistical interactions (Table 1) between treatment factors was added to the manuscript.

Line 216 – Please consider shortening the Discussion section. It is too long…

The Discussion section was revised.

Line 255 (Data) - The authors need to indicate the number of assays from which the results were obtained and to calculate the means and SD (SE) for all parameters

The experiments were performed in triplicate and in three different occasions (n=12). The means and standard deviation were showed in Tables 2 and 3.

Line 259 - Please clarify. This is not clearly understood...

In the Table 2 and 3 the uppercase letters should be matched against uppercase letters on the same line.  Like this, we compare each group in the evaluated times. Lowercase letters should be compared only with lowercase letters in the same column, allowing comparison between groups at each evaluated time.

Line 305 – 309 - This seems unrelated to the study

            Thank you. The Discussion was reviewed and this sentence was removed in the revised Discussion.

Line 303 – 315 - Re-write to 

            Thank you. The Discussion was reviewed and this sentence was removed in the revised Discussion.

Line 319 – Consider revising

It was rewriting. “The extracellular matrix’s construction produces a matrix that…”. The sentence was removed because the extracellular matrix concept was repeating itself.

Line 336 - correct to english as mannans

Thank you for the input. The manuscript was revised.

Reviewer 4 Report

The manuscript by Abreu-Pereira and co-workers is very interesting and well planned. However there are some issues that need to be clarified and corrected in the manuscript, as follows:

Line 48 – Neither ref. 9 or ref. 10 show that beta glucan in the extracellular matrix prevents antifungal activity

Line 82 – Instead of “Reactivation” use the word “Culture”

Line 87 – Why using chloramphenicol to grow candida? It is well known that chloramphenicol acts negatively in yeasts! See these publications

DOI: 10.1016/j.mycmed.2014.10.019

doi:https://doi.org/10.1167/iovs.17-22047

Line 158 – How did you determine biomass? Only by dry weight?

Line 175 - In the Results section please indicate the figure and Table where it can be found the results. 

Line 179 and throughout the entire document – use “.” instead of “,” to indicate decimal place

Line 197 and remaining Figures – the figure caption should have a title instead of “Mean values and …”

Line 200 – What’s the meaning of “MEC's components”?

Line 209 – It’s better to include a table with the statistical study regarding these relations between the period of exposure and the other variables. It is too confusing to read the table as it is…

Line 216 – Please consider shortening the Discussion section. It is too long…

Line 255 (Data) - The authors need to indicate the number of assays from which the results were obtained and to calculate the means and SD (SE) for all parameters

Line 259 - Please clarify. This is not clearly understood...

As said, the authors should make a separate table with the statistical data about these comparisons

Line 305 – 309 - This seems unrelated to the tudy

Line 303 – 315 - Re-write to 

“C. albicans biofilms are resistant to antifungals also due the ECM protector effect, which prevents penetration of antifungal drugs, making their inactivation more difficult”

Line 319 – Consider revising

“The extracel-319 lular matrix’s construction produces a matrix that…”

Line 336 - correct to english as mannans

Author Response

Reviewer 4 comments and suggestions for Authors:

1)The Introduction and Methods are clearly described. Both a susceptible (CAS) and resistant (CAR) strain to nystatin are tested with and without ZER. But why was Nystatin chosen? Candida vaginitis is usually treated with an azole, so it would seem that those experiments, as described in Figure 1 should be tried with fluconazole. For both strains, colony counts are greatly reduced but only at a concentration of 256ug/ml of nystatin

Thank you for the input. The C. albicans ATCC 90028 is susceptible to the fluconazole and the C. albicans ATCC 96901 is fluconazole resistant. According previous study from our group, it was observed that fluconazole-resistant (ATCC 96901) and -susceptible C. albicans (ATCC 90028) strains presented fluconazole MIC values of 256 μg/mL and 16 μg/mL, respectively (Panariello et al., 2018).

The MIC values of nystatin was added to the manuscript by suggestion of Reviewer in the first review round.  Nystatin was only used as an antifungal reference drug to the MIC test and was not used in the treatments in the present study. Topical antifungal agents, such as nystatin is recommended typically as the first-line treatment for uncomplicated cases of oral candidiasis (Akpan & Morgan, 2002).

Panariello BHD, Klein MI, Mima EGO, Pavarina AC. Fluconazole impacts the extracellular matrix of fluconazole-susceptible and -resistant Candida albicans and Candida glabrata biofilms. J Oral Microbiol. 2018 Jun 4;10(1):1476644. doi: 10.1080/20002297.2018.1476644. PMID: 29887974; PMCID: PMC5990947.

Akpan A, Morgan R. Oral candidiasis. Postgrad Med J. 2002 Aug;78(922):455-9. doi: 10.1136/pmj.78.922.455. PMID: 12185216; PMCID: PMC1742467.

2)The data in Table 1 is too complex, especially since a number of features of the organism are tested. Table 1 legend should include a summary of 2-3 sentences which clearly state results. Or the data should be presented per assay type, but separately.

Thank you for the suggestion. The Tables legends were modified and a new Table (Table 1) was added to better visualization of the data.

3) Back to Figure 1. For the CAS strain, the nystatin/ ZER data and the data with the CAR strain, are both not additive. There is a slight difference in the activity of the ZER in combination with nystatin, which I do not understand. If cells are resistant, why would the authors expect a contribution of nystatin in the overall damage to cells?

            The Candida albicans strain ATCC 90028 (CaS) is susceptible to the fluconazole and the strain ATCC 96901 (CaR) is fluconazole resistant. The antifungal Nystatin was only used as an antifungal reference drug in the MIC test. Here, the treatment of CaS and CaR biofilms was performed only with ZER at the concentrations of 128 and 256 µg/mL (according to MIC values observed). The effect of ZER was evaluated in three different times (5 minutes, 10 minutes and 20 minutes) and the biofilm and extracellular matrix components were evaluated.

  1. The experiments of Figure 1 need to be tested for in vitro toxicity of ZER at the concentrations using a mammalian cell line.

In a previous study, it was evaluated in vitro toxicity of ZER at the concentrations of 25, 50 and 100 μg/mL in a mammalian cell line (Moreira da Silva et al., 2018). The results demonstrated that the ZER had no considerable cytotoxicity effect up to 100 μg/mL. At the concentrations 25, 50 and 100 μg/mL the percentages of cell viability were 100, 97 and 92%, respectively, after 24 hour of treatment. The cell viability slightly changed at concentration 50 μg/mL and 100 μg/mL after 48 h treatment. Even after 48 h of ZER treatment, no substantial toxic effect was observed, other than a low cytotoxicity evidenced at 100 μg/mL concentration (reduction of 13% of cell proliferation). According to the authors, the results clearly show that the ZER has no cytotoxic effect on normal mammalian cells at the concentrations evaluated (Moreira da Silva et al., 2018). In the present study, we performed the treatment using the concentration of 128 and 256 μg/mL for only 5 minutes, 10 minutes and 20 minutes. The concentration used in the present study was slightly higher, however the application time was very short.

Moreira da Silva T, Pinheiro CD, Puccinelli Orlandi P, Pinheiro CC, Soares Pontes G. Zerumbone from Zingiber zerumbet (L.) smith: a potential prophylactic and therapeutic agent against the cariogenic bacterium Streptococcus mutans. BMC Complement Altern Med. 2018 Nov 13;18(1):301. doi: 10.1186/s12906-018-2360-0. PMID: 30424764; PMCID: PMC6234655.

  1. Are the concentrations of ZER used in this study like other systems? I worry about toxicity issues of ZEM. See comment #4 above.

Zerumbone is a multifunctional compound with antimicrobial, antitumor, hyperalgesic, antioxidant and anti-inflammatory applications. In a previous study (Kim et al., 2009), animals were fed with a diet containing 100, 250 or 500 ppm ZER (corresponding to 100, 250 and 500 μg/mL) for treatment colon and lung carcinogenesis. Oral administration of ZER resulted in inhibition of proliferation, induction of apoptosis, and suppression in tumors developed in both tissues. It was not observed by the authors any clinical signs of toxicity in the experiment when ZER was added in the diet of the mice. These findings suggest that ZER not promotes cytotoxicity effects.

Kim M, Miyamoto S, Yasui Y, Oyama T, Murakami A, Tanaka T. Zerumbone, a tropical ginger sesquiterpene, inhibits colon and lung carcinogenesis in mice. Int J Cancer. 2009 Jan 15;124(2):264-71. doi: 10.1002/ijc.23923. PMID: 19003968.

  1. The changes in all cell parameters caused by ZER (insoluble biomass, WSP, etc) indicate non-specificity.

            Here, the influence of ZER on biofilm and extracellular matrix components were evaluated. The ZER has been widely used for many studies due to its exceptional biomedical applications. The following are some of the mechanisms of action that have been studied: induction of apoptosis, inhibition of cell proliferation, immune system modulation, activation of cell signaling pathways and antioxidant properties. In summary, ZER has several mechanisms of action that may contribute to its beneficial health effects, including inhibition of fungal proliferation and antioxidant properties.

Shaikh, M. F. et al. (2019). Zerumbone, a bioactive sesquiterpene, modulates multiple pro-inflammatory signaling cascades and inhibits colitis-associated cancer in AOM/DSS mice. Biomedicine & Pharmacotherapy, 109, 975-985.

Lee, Y. H. et al. (2019). Zerumbone suppresses osteopontin expression and breast cancer cell migration, invasion, and metastasis. Molecules, 24(19), 3592.

Thoppil, R. J. et al. (2015). Zerumbone inhibits osteosarcoma cell proliferation and migration via NF-κB and JNK/ERK/MAPK signaling pathways. Journal of cellular physiology, 230(6), 1381-1389.

Chen, C. Y. et al. (2018). Zerumbone from Zingiber zerumbet inhibits innate and adaptive immune responses in Balb/C mice. International immunopharmacology, 55, 312-319.

Round 2

Reviewer 1 Report

Still, both used strains are of a different isogenic line and should not be compared. Despite the fact, that some drug resistance mechanisms are present in the resistant strain, it was not delivered from the sensitive one. The strains might (and most likely are) be dissimilar in basic physiology, which in turn might affect resistance towards the compound. Not only the prescence or the amount of MDR transporters or cyp51 might affect tolerance towards xenobiotics. Some basic physiology, such as permeability of the membrane, thickness of the cell wall, the presence of carbohydrate antiporters, even the ploidity (albicans isolates differ in ploidity - some strains display 3n, or 4n in terms of partial chromosome fragments) might affect the sensitivity towards the xenobiotics. And between strains of the same species such differences occur on the agenda.

If you use two different sets of isogenic lines (lets call them A and B), disturb the same gene (lets call it c) - sensitivty towards the same xenobiotic might differ between A and Bc, similarly as between Ac and B - thus, the most scientifically reliable control would be using A towards Ac, and B towards Bc.

Author Response

Thanks for the comments and clarifications. All comparisons between the strains were  removed from the discussion. I hope that is suitable.

Reviewer 2 Report

In this revised form of the paper entitled "Zerumbone disturbs the extracellular matrix of fluconazole-resistant Candida albicans biofilms", the authors have made improvements regarding the English language Editing requirement.

However, I still believe that MIC values should be provided and in vitro/vivo/silico should be in Italic face. Also, there is no Discussion section (or Results should be Results and Discussion).

Author Response

Thank you for the input. The manuscript was improved as suggested.

The MIC50 values of ZER observed in the susceptibility test were 64 µg/mL for CaR and 256 µg/mL for CaS. It was observed that the OD492nm mean value of the growth control for CaS was 0.756±0.081 and for CaR was 0.686±0.081. It could be observed that the concentration of 256 μg/mL was the lowest concentration of ZER that promoted a 50% reduction (0.318±0.062) in CaS. The concentration of 64 μg/mL promoted a 50% reduction (0.301±0.036) in CaR. For the positive control (nystatin), the MIC values observed were 8 μg/mL (OD492nm0.167±0.062) for CaS and 512 μg/mL (OD492nm = 0.249±0.002) for CaR.

The words were corrected and written in Italic face.  The Material and Method section is separate from the Discussion section.

Reviewer 3 Report

The authors have answered a number of questions raised by this reviewer.  I have no further changes to add.